# Intact Transition Epitope Mapping—Force Differences between Original and Unusual Residues (ITEM-FOUR)

**DOI:** 10.3390/biom13010187

**Published:** 2023-01-16

**Authors:** Claudia Röwer, Christian Ortmann, Andrei Neamtu, Reham F. El-Kased, Michael O. Glocker

**Affiliations:** 1Proteome Center Rostock, University Medicine Rostock and University of Rostock, Schillingallee 69, 18059 Rostock, Germany; 2Waters GmbH|TA Instruments, Helfmann-Park 10, 65760 Eschborn, Germany; 3TRANSCEND Centre–Regional Institute of Oncology (IRO) Iasi, Str. General Henri Mathias Berthelot, Nr. 2–4, 700483 Iași, Romania; 4Department of Microbiology&Immunology, Faculty of Pharmacy, The British University in Egypt, Suez Rd, EL Sherouk City 11837, Egypt

**Keywords:** ITEM-FOUR, nanoESI mass spectrometry, immune complex analysis, personalized genomics, single amino acid polymorphism

## Abstract

Antibody-based point-of-care diagnostics have become indispensable for modern medicine. In-depth analysis of antibody recognition mechanisms is the key to tailoring the accuracy and precision of test results, which themselves are crucial for targeted and personalized therapy. A rapid and robust method is desired by which binding strengths between antigens and antibodies of concern can be fine-mapped with amino acid residue resolution to examine the assumedly serious effects of single amino acid polymorphisms on insufficiencies of antibody-based detection capabilities of, e.g., life-threatening conditions such as myocardial infarction. The experimental ITEM-FOUR approach makes use of modern mass spectrometry instrumentation to investigate intact immune complexes in the gas phase. ITEM-FOUR together with molecular dynamics simulations, enables the determination of the influences of individually exchanged amino acid residues within a defined epitope on an immune complex’s binding strength. Wild-type and mutated epitope peptides were ranked according to their experimentally determined dissociation enthalpies relative to each other, thereby revealing which single amino acid polymorphism caused weakened, impaired, and even abolished antibody binding. Investigating a diagnostically relevant human cardiac Troponin I epitope for which seven nonsynonymous single nucleotide polymorphisms are known to exist in the human population tackles a medically relevant but hitherto unsolved problem of current antibody-based point-of-care diagnostics.

## 1. Introduction

Accurate disease diagnosis, which is in many cases assisted by precise laboratory results, is a prerequisite of evidence-based medicine [1]. Moreover, targeted care and pinpointed therapy would not have been possible without advanced molecular diagnostics, paving the way for the era of “personalized medicine” [2]. Diagnostic assays that are based on antibodies and their extraordinary molecular recognition specificities have become of utmost importance, e.g., in point-of-care testing of an individual’s diseases, including insults to health conditions such as acute myocardial infarction (AMI) [3]. AMI diagnostics has been optimized to determine with great certainty minute amounts of heart muscle proteins, such as human cardiac troponin T (hcTn T) and/or human cardiac troponin I (hcTn I), which have been released from raptured heart tissue into an individual´s peripheral blood, e.g., upon cardiac necrosis [4]. During assay development and optimization enormous efforts were undertaken to target only such epitopes with monoclonal antibodies and therefrom derived derivatives e.g. single-chain antibodies [5] which (i) differentiate between cardiac and skeletal troponin proteins, i.e. to avoid cross-reactivity [6], (ii) stay freely accessible on the respective target proteins as opposed to such regions which may be (partially) protected due to protein-protein interactions in circulating protein complexes [7], (iii) are not altered by (sub-stoichiometric) post-translational modifications [8], and (iv) remain stably detectable on circulating antigens during an appropriate time window which is requested for blood collection from the suspected attack-suffering patient [9].

By contrast, the role of an epitope´s amino acid sequence variations on antibody recognition, which in a population exist because of coding single nucleotide polymorphisms (SNPs) of the respective protein sequence, has so far not been systematically investigated, despite the fact that coding SNPs are by far the most common reasons for single amino acid polymorphism (SAP) manifestations in proteins [10]. SAPs seem of particular and immediate medical importance in the case of AMI diagnostics, and accordingly, respective concerns have been raised, suspecting that SAPs were at least partially responsible for the unsatisfactory sensitivity and specificity of commercial AMI diagnostic assays [11]. Fears have been articulated that TnT and/or TnI epitope SAP carriers who belong to the increased cardiomyopathy risk group [12] are in deadly peril when suffering from a heart attack due to potentially being misdiagnosed and eventually mistreated because of false negative cardio-specific tests.

With our newly developed mass spectrometric epitope mapping procedure, “Intact Transition Epitope Mapping (ITEM)” or “Intact Transition Epitope Mapping—One-Step Non-Covalent Force Exploration (ITEM-ONE)”, we have shown rapid identification of epitope regions on protein surfaces that encompass partial surfaces typically made of about 7 to 15 amino acid residues [13]. In addition, antibody-epitope peptide binding strengths and dissociation energies have been determined and ranked by us applying our gas phase epitope mapping method, “Intact Transition Epitope Mapping—Thermodynamic Weak-force Order (ITEM-TWO)” [14,15]. A comparison to in-solution K_D_ values of immune complexes showed that the gas phase-originated rankings matched fairly well with those obtained by solution-based measurements [14,16]. Furthermore, the method “Intact Transition Epitope Mapping—Targeted High-Energy Rupture of Extracted Epitopes (ITEM-THREE)” developed by us has demonstrated its ability to precisely determine unknown epitope peptides via mass spectrometric amino acid sequencing of antigenic determinants upon release of a respective epitope peptide from its immune complex [16,17]. Our developed mass spectrometry-based epitope mapping methods are fast, consume only picomoles of biological material, and provide results with excellent resolution and high accuracy [18,19]. Our previous work on epitope mapping [20] has thus laid the groundwork for precise investigations that are requested to study the effects of SAP-derived epitope sequence alterations on antibody binding in all their detail.

The monoclonal anti-hcTn I antibody (clone MF4) has been picked for this proof-of-principle study because it has been considered a candidate for point-of-care diagnostics by several groups [7,21,22,23]. The epitope of this antibody has been assigned to the mobile C-terminal region of hcTn I, more precisely to the partial sequence GDWRKNID (aa189–aa196) [6,24]. The eight hcTn I 15-mer peptides investigated here cover the partial amino acid sequence range aa184–aa198 from the C-terminal region of hcTn I. In addition to the wild-type amino acid sequence, seven peptides represent SNP-related exchanges of single amino acid residues at positions aa186, aa192, and aa193. These single amino acid polymorphisms (SAPs) are correlated with increased risks of cardiomyopathies [25,26,27,28,29,30,31,32]. Epitope peptides were ranked according to their experimentally determined dissociation enthalpies relative to that of the wild-type epitope peptide, thereby revealing which SAPs caused weakened, impaired, and even abolished antibody binding. The procedure described here, which combines personalized genomics with cutting-edge mass spectrometric immune complex analysis, can be used to investigate antibody—antigen interactions with single amino acid residue resolution in general. We therefore termed this targeted experimental approach of binding motif characterization “Intact Transition Epitope Mapping—Force differences between Original and Unusual Residues (ITEM- FOUR)”.

## 2. Materials and Methods

### 2.1. Preparation of Antibody, Peptide, and Immune Complex Solutions

Starting materials and antibody-peptide complex-containing solutions were prepared as previously described [19]. In brief, an aliquot of 25 µL (50 µg) of the monoclonal anti-hcTn I antibody [MF4] (ab38210 from Abcam, Cambridge, United Kingdom) was placed on a 10K centrifugal filter device (Merck Millipore, Carrigtwohill, Ireland) with 400 µL 200 mM ammonium acetate, pH 6.7. After centrifugation at 13,000 rpm for 7 min in an Eppendorf centrifuge (MiniSpin, Eppendorf, Hamburg, Germany), the flow through was discarded and the filter was filled-up with 450 µL 200 mM ammonium acetate, pH 6.7. This centrifugation/discarding/refilling step was repeated eight times. Afterwards, the filter was placed upside down in a new tube and centrifuged for 5 min at 4500 rpm. Approximately 50 µL of antibody solution were collected (solution 1). The protein concentration was determined using the Qubit^TM^ 2.0 Fluorometer (Invitrogen by Life Technologies/Thermo Fisher Scientific, Waltham, MA, USA) assay and revealed 0.35 to 0.25 mg/mL. Synthesized hcTn I peptides (Table 1) were purchased as lyophilized powders (peptides and elephants, Hennigsdorf, Germany) and dissolved in 200 mM ammonium acetate, pH 6.7, to obtain peptide concentrations of ca. 2.0 mg/mL, each. The actual peptide concentrations of individual peptide solutions were determined using the Qubit^TM^ 2.0 Fluorometer assay. Afterwards, all peptide solutions were diluted 1:100 with 200 mM ammonium acetate, pH 6.7 (solution 2). To obtain immune-complex-containing mixtures, the antibody solution (solution 1) and the peptide solution (solution 2) were mixed to obtain final molar antibody-peptide ratios of 1:2.2 (solution 3). Typically, 4 µL of antibody solution were mixed with the appropriate volume (typically 1–2 µL) of the individual peptide solution. Solutions 3 were incubated at room temperature for at least 1 h.

### 2.2. Offline Nanoesi-MS Instrument Settings and Data Acquisition

For each ITEM-FOUR examination, 3 µL of the peptide, antibody, or immune complex-containing solution were loaded into nanoESI capillaries that were pulled and gold coated in house [14]. NanoESI-MS measurements were performed with a Synapt G2-S mass spectrometer (Waters MS-Technologies, Manchester, United Kingdom) with the following settings for ITEM-FOUR measurements: capillary voltage, 1.8 kV; source temperature, 40 °C; source offset voltage, 110 V; sample cone voltage, 110 V; cone gas flow, 100 L/h; trap gas flow, 8.0 mL/min; transfer collision cell voltage, 2 V. Measurements were acquired in positive-ion resolution mode, applying a mass window of m/z 200–8000. This mass range was calibrated using 1 mg/mL sodium iodide dissolved in isopropanol/water (50:50, *v*/*v*). The quadrupole analyzer was used with the following settings: M1 = 4000, dwell time, and ramp time 25%; M2 = 5000, dwell time, and ramp time 25%; and M3 = 6000 to block transmission of lower molecular weight ions. The first collision cell (TRAP) was used to dissociate antibody-peptide complexes by increasing the collision cell voltage difference (∆CV) in a stepwise manner every 60 s. The collision cell voltage difference steps were: 2, 4, 6, 8, 12, 16, 20, 30, 40, 50, 60, 70, and 80 V. Measurements were repeated at least twice (up to seven times) per immune complex, generating a total of 32 measurement series. The mass spectrometry proteomics data have been deposited to the ProteomeXchange Consortium via the PRIDE [33] partner repository with the dataset identifier PXD038365.

### 2.3. Offline NanoESI-MS Data Analysis

MassLynx version 4.1 (Waters MS-Technologies) was used for data analysis as described previously [19]. First, the scans of each ∆CV setting encompassing 60 s of recording were combined to generate an average mass spectrum for each ∆CV step. Next, the mass spectra were smoothened in 20 cycles with a window of 10 using the Savitzky-Golay algorithm. The ion intensities of all molecular species and their corresponding charge states (complex ions, complex-released peptides, and antibody fragments) were read after smoothing. A bell-shaped Gaussian curve of the intensities of each molecular species was calculated using Origin 2017 (32-bit) SR2 (OriginLab Corporation, Northampton, MA, USA). Missing values had to be imputed because the Gauss fit calculation requires at least 5 values. Imputed values were chosen with higher and lower molecular weights, respectively, and their intensities were set to the intensity of the background at the respective m/z range. The apex values of the Gaussian fits were used to calculate the normalized intensities of educts (height of antibody-peptide complex with 1 peptide + height of antibody-peptide complex with 2 peptides/∑ of heights of all molecular species). Furthermore, the mean charge states of the multiply charged ion series were calculated using the m/z values of the apices of the fitted Gaussian curves.

### 2.4. ITEM-FOUR Analysis of Apparent Kinetic and Apparent Thermodynamic Values

Calculations of apparent kinetic and thermodynamic values followed published protocols [16,34]. The measurement series went into further analysis when it met at least three of the four criteria: (i) The ion signals of the antibody-peptide complex enable differentiation of the ion signals of antibodies with no peptide, one peptide, and two peptides, respectively, in at least five charge states, i.e., the minimum difference between the corresponding ion signals is at least 50%; (ii) no presence of sodium adduct ions (if the intensity of the doubly charged sodium adduct ion signal was higher than 35% of the doubly charged protonated peptide ion signal intensity, which was determined at ∆CV = 80 V after smoothing, the entire measurement series was excluded from further analysis); (iii) no outlier values have to be excluded for Boltzmann fitting; (iv) the regression coefficient R2 of the Boltzmann fit is ≥0.99. Normalized intensities of educts (mean values and standard deviations of replicate measurements) were plotted against the ∆CV values for each measurement, and a Boltzmann curve was fitted to the data points using the Origin software. Boltzmann curve parameters (initial values A1, final values A2, centers x0, and course constants dx) of fits were recorded and used for calculating the equations of the tangent lines along the steep parts of the Boltzmann curves. The mathematical procedures for calculations of kinetic and thermodynamic values using the Eyring-Polanyi equation, the Arrhenius equation, the Gibbs-Helmholtz equation, and the van´t Hoff equation have been followed as published [16,34]. As the amino acid sequence of the anti-hcTn I antibody used in this experiment is not known, the number of atoms for the antibody was set to 20,000, according to the calculated number of atoms from rituximab, an antibody with a well-known sequence and the same isotype (IgG1).

### 2.5. Isothermal Titration Calorimetry in Solution

The monoclonal anti-hcTn I antibody [MF4] (ab38210 from Abcam, Cambridge, United Kingdom) was stored in PBS (0.138 M NaCl; 0.0027 M KCl, pH 7.4 from Sigma-Aldrich, St. Louis, MO, US), pH 7.4, and was used as purchased. The hcTn I peptides 1, 2, 4, 6, 7, and 8 were dissolved in PBS and stock solutions with 0.32 to 0.93 mg/mL peptide concentrations were prepared.

Isothermal titration calorimetry measurements were performed with an Affinity ITC instrument (Waters|TA Instruments, New Castle, DE, USA) at T = 298 K equipped with a 190 µL sample cell and a matching reference cell, AccuShot for precise delivery of the titrant, and a Flex Spin for slow speed stirring (125 rpm), efficient mixing, and highest sensitivity. The anti-hcTn I antibody was placed in the sample cell at 1.95 µM (in PBS). The hcTn I peptides, each at 40 µM (in PBS), were automatically titrated to the antibody in 18–20 steps (2 µL per step). The time between titrations was 200 s to establish a baseline between each injection for accurate heat determination and modeling.

The integrated areas under each injection peak (heat over time) were plotted against the active species’ respective molar ratios ranging from 1:0.4 to 1:6.6. An independent binding model was chosen to fit all the data and directly determine the enthalpy (ΔH) and association constant (K_A_). Gibbs free energy (ΔG) is calculated from the K_A_ using the van´t Hoff equation. The change in entropy term (ΔS) is calculated applying the Gibbs-Helmholtz equation.

### 2.6. Atomistic Molecular Dynamics Simulations

The PepFold3 (https://bioserv.rpbs.univ-paris-diderot.fr/services/PEP-FOLD3/; accessed on 19 October 2022) predicted conformations of the hcTn I peptides (see Table 1) were used as initial structures for atomistic molecular dynamics simulations. The protonation states of the simulated peptides were chosen based on their pIs and the pH that had been used in the ITC measurements (Appendix A). Each peptide was in-silico solvated in a cubic box of water, considering a minimum distance of 10 Å from the box´s edges, so that the peptides did not interact with their periodic images. Ions were added to neutralize the systems. After an equilibration stage, the peptides were simulated for 50 ns at constant temperature and pressure. The CHARMM27 [35] force field was used for peptides, and the special CHARMM-TIP3P model was selected for water. All simulations and analysis were performed with the GROMACS-2019 [36] software suite. The Chimera 1.14rc (UCSF) [37] software was used to visualize the structures. The solvent-accessible surface areas (SASAs) of the epitope peptides were calculated before and after a 50 ns simulation using the “EpiMED-Surf” web tool (http://www.pzr.uni-rostock.de/Surfacer/) [38]. SASAs were computed for each atom (excluding H atoms) using the “rolling ball” or Shrake Rupley algorithm [39]. A probe sphere with 960 points was rolled along the van der Waals surfaces; its center depicts the SASA. The pdb files of the epitope peptide structures before and after simulations were used as input files. RMSD calculations were performed as described elsewhere [19,36]. In brief, the root mean square deviation (RMDSD) of atom distances was calculated using the “rmsdist” utility in the GROMACS 2019 suite. The reference structure was taken from the input structure file. RMSD calculations were performed as described elsewhere [19,36]. In brief, the root mean square deviation (RMDSD) of atom distances was calculated using the “rmsdist” utility in the GROMACS 2019 suite. The reference structure was taken from the input structure file. The RMSD was calculated at a given time ‘t’ as the root mean square of the differences in distance between the structure’s atom-pairs at that time ‘t’ in relation to the reference structure.

## 3. Results

### 3.1. Mass Spectrometric Characterization of hcTn I Epitope Peptides and Anti-hcTn I Antibody

The consensus hcTn I epitope motif of a commercially available monoclonal anti-hcTn I antibody [MF4] is represented by a synthetic 15-mer peptide (wild type). Seven mutated hcTn I epitope peptide derivatives reflect known SAPs within or near the epitope region (Table 1 and Appendix A).

**Table 1 biomolecules-13-00187-t001:** Single nucleotide polymorphism information and peptide masses for the anti-hcTn I antibody [MF4] epitope.

Peptide No. (wt or SAP) ^(a)^	Amino Acid Sequence (wt or SAP) ^(a)^	SNP Entry ^(b)^	SNP ^(b)^	Cardiomyopathy Association ^(b)^	Calc. m/z (Charge)	Exp. m/z (Charge)
**1 (wt)**	ENREVGDWRKNIDAL	n.a.	n.a.	n.a.	605.64 (3+)	605.63 (3+)
**2 (R186Q)**	EN**Q**EVGDWRKNIDAL	rs397516357	G > A	hypertrophic	596.30 (3+)	596.30 (3+)
**3 (R192H)**	ENREVGDW**H**KNIDAL	rs104894729	G > A	hypertrophic/restrictive	599.30 (3+)	599.31 (3+)
**4 (R192L)**	ENREVGDW**L**KNIDAL	rs104894729	G > T	hypertrophic	591.30 (3+)	591.32 (3+)
**5 (R192C)**	ENREVGDW**C**KNIDAL	rs727503499	C > T	restrictive/hypertrophic	587.95 (3+)	587.96 (3+)
**6 (D190G)**	ENREVG**G**WRKNIDAL	rs104894728	A > G	familial hypertrophic	586.31 (3+)	586.32 (3+)
**7 (R192P)**	ENREVGDW**P**KNIDAL	rs104894729	G > C	hypertrophic	585.96 (3+)	585.98 (3+)
**8****(R192P)** **8** **(K193E)**	ENREVGDW**P**ENIDAL	rs104894729 rs730881080	G > C A > G	hypertrophic hypertrophic	878.91 (2+)	878.94 (2+)

^(a)^ aa184–aa198 from hcTn I (UniProt: P19429); the epitope region of the monoclonal anti-hcTn I antibody (clone MF4, ab38210 from Abcam) is underlined; amino acid exchanges in peptides 2–8 are printed in bold and are colored; wt: wild type; SAP: single amino acid polymorphism. ^(b)^ SNP: single nucleotide polymorphism; https://www.ncbi.nlm.nih.gov/snp/ (accessed on 14 October 2022); n.a.: not applicable.

The purities and correct amino acid compositions of the eight hcTn I epitope peptides were verified by offline nanoESI-MS measurements (Appendix A). The experimentally determined molecular masses matched the respective calculated masses (Table 1). Additionally, MS/MS fragmentation analyses confirmed the correct amino acid sequences of the hcTn I epitope peptides (data not shown).

Likewise, the purity of the monoclonal anti-hcTn I antibody solution after solvent exchange was checked by offline nanoESI-MS analysis which showed solely one group of baseline separated multiply charged ion signals (from 23+ to 27+) between m/z 5000 and m/z 7000 with intensities following a Gaussian distribution (Appendix A). An average molecular mass of 146,756.2 (±70) Da was calculated from three measurements. This mass is similar to the mass of rituximab (average molecular mass 147,457.9 (±46) Da; Appendix A), for which the number of atoms is approx. 20,000.

### 3.2. Mass Spectrometric Analysis of hcTn I Epitope Peptide Binding to the Anti-hcTn I Antibody

For gas-phase binding strength analysis of immune complexes, the transmission of low-mass ions (m/z 200–m/z 3000) was blocked using the mass spectrometer´s first mass filter (quadrupole mass analyzer). Immune-complex dissociation occurred in the collision cell by applying increased collision cell voltage differences. After leaving the collision cell and before reaching the detector, ions with different masses were separated by the instrument´s second mass filter (ToF mass analyzer).

The mass spectra of the mixture with epitope peptide 1 (wild-type) and the anti-hcTn I antibody (mixed in solution in a 2.2:1 molar ratio) showed intensely multiply charged antibody and immune complex ions between m/z 5000 and m/z 7000 at a low collision cell voltage difference (Figure 1a). The instrument´s resolution enabled it to differentiate multiply charged antibody ion signals (24+ to 27+) from ion signals that originated from antibody-peptide immune complexes that had bound one and two peptides, respectively. The ion signal of the immune complex with one bound peptide was the most intense. As collision energies increased, the intensity of doubly and triply charged isotope-resolved ion signals in the low-mass ranges of the mass spectra increased (Figure 1b–d). The experimentally determined masses of the peptide ions, which had been released from the immune complexes as protonated ions, exactly matched the calculated mass of epitope peptide 1.

From all series of mass spectra, the ion signal intensities of all ion species (antibody-peptide complex ions, complex released peptide ions, antibody ions, and antibody fragment ions) were recorded (Appendix A), and bell-shaped curves with Gaussian distributions were fitted onto the intensity values of the respective ion series to obtain their corresponding apex heights (Appendix A). Both the mean charge states (Appendix A) and the normalized educt ion intensities were determined from the apex heights. Decreases of educt ion intensities, i.e., of the immune complex ions with one and two bound peptides as functions of collision cell voltage differences, showed sigmoidal-shaped courses (Figure 2). The curve was fitted to the mean from repetitive measurements using a Boltzmann function, and fitting accuracy was greater than 0.99.

Analyses of immune complexes of the anti-hcTn I antibody upon incubation with the mutated hcTn I epitope peptides 2 (R186Q) and 6 (D190G) showed similar results (Appendix A). The ion signal intensities that corresponded to the antibody-peptide complexes with one peptide were higher than the ion signals for the antibody with two bound peptides or the antibody ion signals alone. Increasing collision cell energies led again to dissociation of the immune complexes, which caused intensity increases of doubly and triply charged peptide ion signals (Appendix A). When compared to the complex dissociation of the hcTn I wild-type peptide 1, the intensity courses and slopes were very similar (Figure 2).

Likewise, the mutated hcTn I peptides 3 (R192H), 4 (R192L), and 5 (R192C) were incubated with the monoclonal anti-hcTn I antibody in molar ratios of 2.2:1, and again, multiply charged ion signals between m/z 5000 and m/z 7000 were detected that allowed the differentiation of the antibody ion signals from the ion signals of the immune complexes with one and two bound peptides, respectively (Figure 3a, Appendix A). However, comparing the ion signal intensities of the different molecular species at low collision cell voltage differences revealed that immune complex compositions with hcTn I peptides 3, 4, and 5 were different from those of hcTn I peptides 1, 2, and 6. In all spectra with hcTn I peptides 3, 4, and 5, the ion signal intensities of the antibody alone were higher than the immune complex ion signals (cf. Figure 3, Appendix A). Nevertheless, increasing the collision cell voltage difference again caused in all cases increases in doubly protonated peptide ion signals, indicating dissociation of peptides 3, 4, and 5, respectively, from the anti-hcTn I antibody (Figure 3, Appendix A). However, dissociation yields were lower than those of peptides 1, 2, and 6.

Again, the ion signal intensities of all ion species were extracted from all series of mass spectra (Appendix A), and bell-shaped curves were fitted to the respective intensity values to calculate normalized educt ion intensities (Appendix A) and mean charges (Appendix A). Boltzmann course functions were fitted to the decreases of educt ion intensities over cell voltage differences (Figure 4), and fit accuracies were greater than 0.99 in all cases (Table 2). The sigmoidal-shaped courses exhibited shallower slopes (−0.57–0.35) as compared to the steeper slopes of the courses for peptides 1, 2, and 6 (−0.67–−0.85). Thus, the binding behavior of peptides 1 to 6 may be summarized by the term “orthodox binding”, which indicates that the required interactions between paratope and epitope are present to yield strong binding despite certain amino acid exchanges within or near the epitope region.

**Table 2 biomolecules-13-00187-t002:** Course characteristics of gas phase dissociations of the immune complexes of hcTn I epitope peptides and anti-hcTn I antibodies.

Peptide No.	Amino Acid Sequence ^(a)^	Initial [%] ^(b,c)^	Final [%] ^(b,d)^	∆CV_50_ [V] ^(e)^	dx [V] ^(e)^	Slope [%/V] ^(e)^	R^2 (e)^
1	ENREVGDWRKNIDAL	77.36	36.24	30.0	15.2	−0.67	0.997
**2**	EN**Q**EVGDWRKNIDAL	87.39	47.63	30.3	11.7	−0.85	0.998
**3**	ENREVGDW**H**KNIDAL	52.55	32.68	27.1	12.3	−0.40	0.997
**4**	ENREVGDW**L**KNIDAL	41.60	23.12	27.2	13.2	−0.35	0.996
**5**	ENREVGDW**C**KNIDAL	58.77	29.41	27.7	13.0	−0.57	0.996
**6**	ENREVG**G**WRKNIDAL	58.83	23.33	32.1	11.8	−0.75	0.997
**7**	ENREVGDW**P**KNIDAL	15.37	5.90	n.a.	n.a.	n.a.	n.a.
**8**	ENREVGDW**P****E**NIDAL	0.00	0.00	n.a.	n.a.	n.a.	n.a.

^(a)^ aa184–aa198 from hcTn I (UniProt: P19429); the epitope region of the monoclonal anti-hcTn I antibody (clone MF4, ab38210 from Abcam) is underlined; amino acid exchanges in peptides 2–8 are printed in bold and are colored. ^(b)^ averaged amounts at the corresponding ∆CV from all measurements (cf. Figure 2 and Figure 4). ^(c)^ the immune complex amount at the lowest applied ∆CV (2 V). ^(d)^ the immune complex amount at the highest applied ∆CV (80 V). ^(e)^ n.a. = not applicable.

In contrast to the orthodox binding behavior of peptides 1 to 6, the mixture, which consisted of peptide 7 (R192P) and the monoclonal anti-hcTn I antibody, showed multiply charged ion signals between m/z 5000 and m/z 7000, which corresponded to free antibody, but the typical satellite ion signals of immune complexes were seen with only very low intensities (Appendix A). Moreover, the doubly protonated peptide ion signal of peptide 7 was detected only in low abundances despite increasing collision cell voltage differences (Appendix A), and the course of educt ion intensities as a function of collision cell voltage differences remained flat (Figure 4), indicating that peptide 7 was bound by the paratope in an unorthodox fashion. Proof of unorthodox binding, as opposed to unspecific binding of peptide 7 to the anti-hcTn I antibody at a paratope-different surface area, was obtained by mixing peptide 7 with the unrelated anti-Histag antibody. With this binary mixture, no complex formation was detected (Appendix A).

At last, mass spectrometric ITEM analysis of the mixture that contained peptide 8 (R192P, K193E) and the monoclonal anti-hcTn I antibody revealed multiply charged ion signals between m/z 5000 and m/z 7000 that corresponded to the antibody alone. Satellite ion signals were completely missing (Appendix A). Consequently, no peptide ion signals were detected in the low mass ranges of the mass spectra (Appendix A) despite increasing collision cell voltage differences. Note that transmission of low-mass ions from the source into the collision cell had been blocked. These results confirmed that peptide 8 did not bind at all to the monoclonal anti-hcTn I antibody.

### 3.3. ITEM-FOUR Analysis of Immune Complex Dissociation in the Gas Phase

From the curve parameters of the Boltzmann fits to the courses of educt ion intensities as functions of collision cell voltage differences (Table 2), apparent thermodynamic and apparent kinetic values of gas-phase complex binding strengths were calculated for the orthodox binding peptides 1 to 6 (Table 3). The mathematical procedures for calculations of kinetic and thermodynamic values employed the Eyring-Polanyi equation, the Arrhenius equation, the Gibbs-Helmholtz equation, and the van´t Hoff equation. Extrapolations were applied to reach equivalency with resting and neutral complex dissociation (cf. Appendix A), and the reported apparent values (#) are marked with “m0g”, which refers to the (i) mean charge state (m), (ii) without excess temperature or energy of the immune complex (0), and (iii) determined in the gas phase (g).

In sum, from the monoclonal anti-hcTn I antibody binding peptides 1 to 7, peptides 1 to 6 are orthodox binders, which themselves fall into two distinctive groups according to their ${\Delta H}_{m0g}^{\#}$ values. Peptides 1, 2, and 6 are “very strong binders” (group I) as their ${\Delta H}_{m0g}^{\#}$ values of the gas-phase dissociation reaction are above −2 kJ/mol, and peptides 3, 4, and 5 are grouped as “strong binders” (group II) with ${\Delta H}_{m0g}^{\#}$ values below −2 kJ/mol. As $T_{amb}{\Delta S}_{m0g}^{\#}$ values are almost the same for all six tested cases, there seems to be no relevant entropy contribution for the differing gas-phase dissociation reactions. Peptide 7 is regarded as a “weak and unorthodox binder” (group III), while peptide 8 does not bind at all and therefore belongs to the “non-binder” group (group IV).

### 3.4. Isothermal Titration Calorimetry of Immune Complex Formation in Solution

To compare gas-phase dissociation behaviors to in-solution complex formation reactions of immune complexes consisting of one hcTn I epitope peptide and the anti-hcTn I antibody at a time, we performed isothermal titration calorimetry (ITC) experiments with selected epitope peptides. In-solution complex binding strengths were estimated from their measured dissociation constants, K_D_ (Table 4).

The four group-representing peptides, i.e., peptide 1 (“very strong binders”; group I), peptide 4 (“strong binders”; group II), peptide 7 (“unorthodox and weak binders”; group III), and peptide 8 (“non-binders”; group IV), were ranked according to the ITC results, which showed that peptide 1 bound stronger than peptide 4, and peptide 4 bound stronger than peptides 7 (group III) and 8 (group IV). Peptides 7 and 8 did not bind in solution. Intriguingly, the in-solution binding strength of peptide 2 (Table 4) was consistently ranked as the strongest when compared with the gas phase ranking. Surprisingly, peptide 6, which by ITEM-FOUR was ranked as a “very strong binder” (group I), was determined by ITC to be an even weaker binder than peptide 4 (Table 4). The deviating ranking of peptide 6’s binding strengths uncovered solvation-related dependencies.

### 3.5. Atomistic Molecular Dynamics Simulations of hcTn I Epitope Peptide Structure Changes

Predicted 3D structures for all epitope peptides were obtained from PepFold3 and served as starting points for molecular dynamics simulations that mimicked structural in-solution flexibilities. The epitope peptides´ secondary structure element involvements were examined on the amino acid residue level, probing one of eight secondary structure categories at a time (α-helix; 3_10_-helix, π-helix, turn, bend, β-bridge, β-sheet, and coil) during the 50 ns simulation time periods (Figure 5 and Appendix A).

The structures of peptides 1 and 2, and to a great extent also the structures of peptides 3 and 4, remained fairly stable over the simulation time periods, as is seen from the fairly homogenously distributed pattern of helical conformation contents. Secondary structure categories are represented by a total of 10,000 colored bars per line (Figure 5). The blue bars (helical conformation) dominated each of the lines of the epitope peptides´ residues, which in the wild type (peptide 1) were the amino acid residues VGDWRKN. Border-located amino acid residues ENRE at the N-termini and IDAL at the C-termini of the epitope peptides adopted mostly non-helical secondary structures over the entire simulation time periods. The total helical contents over the entire simulation time period (50 ns) of peptides 1, 2, 3, and 4 were 65%, 60%, 35%, and 50%, respectively (Appendix A). Greater losses of initially higher helical secondary structures over the entire simulation time period were read for epitope peptides 5, 6, 7, and 8, where helical contents initially (first 25 ns) were 37%, 25%, 29%, and 34%, but then (last 25 ns) fell to 17%, 2%, 6%, and 25%, respectively (Appendix A). From them, the helical contents of the epitope regions (wild-type partial amino acid sequence “GDWRKNID”) were maintained longest by peptide 5. By contrast, peptide 8 maintained helical content at partial amino acid sequence VGDW but not at partial amino acid sequence PENID over the 50 ns simulation time period. Interestingly, peptides 6 and 7 kept their helical contents in the epitope core regions RKNI and PKNI, respectively, for about half of the simulation time periods.

Thus, the molecular dynamics simulation results stand in agreement with the experimental data, yet with peptide 6 as the one exception. Together, they point out the importance of adopting and maintaining a mostly helical structure of the epitope peptides in general and particularly in the peptides´ epitope core regions for being bound by the anti-hcTn I antibody. Strongest binders (peptides 1 and 2, group I) remained almost completely helical, and strong binders (peptides 3, 4, and, to some extent, 5, group II) kept helical structures at least in the epitope core regions throughout the simulation time period. Weak binders or non-binders (peptides 7 and 8; groups III and IV) rather quickly abandoned their helical structures, particularly in important epitope core regions.

We increased the simulation time period for epitope peptides 1 and 6 to 200 ns to confirm that the recognition structure of the anti-hcTn I antibody was exposed on the epitope peptide in its α-helical conformation. The secondary structure calculations for these extended simulations clearly showed that peptide 1 remained helical throughout the simulation time, while peptide 6 lost its partial helical conformation fairly rapidly (Appendix A). The RMSD plots for peptides 1 and 6 were calculated from the extended 200 ns simulations. Peptide 1 showed low RMSDs (below 0.2 nm) throughout the entire simulation time period, whereas peptide 6 displayed much larger deviations from the starting structure, which was due to unfolding (Appendix A). The SASA calculation results are in line with the information obtained by the RMSD calculations and agree with maintaining the initial mostly helical structures, at least in part, by peptides 1, 2, and 4, and rather rapid losses of helicities by peptides 3, 5, 6, 7, and 8 (Appendix A).

## 4. Discussion

In solution, the gain of enthalpy from stable complex formation outweighs the sum of two energy penalties: (i) the desolvation enthalpy costs of both free antibody and free ligand, and (ii) the conformational entropy loss caused by rearranging free ligand to adopt a “bound conformation” [40]. For epitope peptides with comparable amino acid compositions, desolvation enthalpies should be nearly the same and, hence, might be negligible for the ranking of binding strengths of related ligands to a given antibody. Similarly, when ligand solution conformations are close to complex-bound conformations, conformational entropy losses may be insignificant. On the contrary, conformational rearrangements prior to or during complex formation become important and can significantly reduce binding affinities. As a result, structural differences in epitope peptides caused by even minor amino acid exchanges may affect binding strengths to the same antibody paratope due to associated conformational rearrangement energy obligations as well as differences in desolvation enthalpies. Note, prolinyl and glycinyl residues—either of them were introduced as SAPs in hcTn I epitope peptides 6, 7, and 8—as well as clusters of these two amino acid residues are secondary structure breakers [41], and their introduction into an epitope region can cause measurable effects on antibody binding strengths.

Ranking of binding strengths of related epitope peptides to a given antibody is possible by classical in solution methods, such as ITC, surface plasmon resonance recording, and other related techniques [20]. But from the binding energy determinations alone, mutual in-solution structural rearrangement contributions remain concealed. The ITEM-4 method, which we developed, allows us to rank immune complex binding strengths with SAP-containing epitope peptides in a fast and efficient manner with consuming only little material. ITEM-FOUR rankings reveal binding strength differences in pure intermolecular interactions since dissociation of immune complexes in the gas phase is per se free of solvation-related processes. Such non-covalent interactions prevail in the gas phase for hundreds of milliseconds [42,43] which is long enough to be investigated by modern mass spectrometry methods. Previously published examples of binding strength ranking comparisons demonstrated that results from ITEM measurements matched well with rankings from in-solution affinity determinations [13,14]. However, the ranking of binding strength in the gas phase deviated from the in-solution ranking in the case of the anti-hcTn I antibody—epitope peptide 6 immune complex. According to molecular dynamics simulations, it seems likely that in solution, peptide 6 underwent substantial conformational changes to adopt the helical-bound ligand conformation. Without comparisons between in-solution and gas-phase rankings, solvent effects, through which solvent-related structural contributions of ligands were added to immune complex affinity, remained hidden.

Structural flexibility as seen with the hcTn I epitope peptides may be present in a partial structure of the peptide-originating antigen protein as well, and this possibility increases particularly when the respective underlying protein region is considered to be (intrinsically) unstructured [44]. In fact, the hcTn I protein seems an example of a “mostly folded, local disorder” protein [45] since its by X-ray determined 3D structure lacked the C-terminal region [46]. Moreover, the C-terminal region of hcTn I has been modeled by alphafold [47] as being unstructured (https://alphafold.ebi.ac.uk/entry/P19429; accessed on 16 September 2022), which stands in agreement with X-ray data. In silico modeling of the SAP-carrying hcTn I epitope peptides investigated here, which represent the C-terminal region of hcTn I, suggested predominantly helical conformations.Pepfold3 is a novel computational framework especially developed for peptide secondary structure prediction, which together with molecular dynamics simulations has been successfully used in the past for structural peptide computational studies [48,49]. We speculate that, on the epitope peptides´ surfaces, the side chains of the respective amino acid residues were displayed in accurate positions for strong antibody interactions when helical structures were adopted by the epitope peptides. Note that “very strong” and “strong” binding peptides (groups I and II) were tagged by molecular dynamics simulations as mostly maintaining helical structures throughout the simulation time periods. Even peptide 6, which displayed its epitope core region in a non-helical conformation for more than half of the simulation time period, adopted a helical conformation for a fraction of total simulation time. The anti-hcTn I antibody´s recognition motif was then present and enabled immune complex formation in an orthodox fashion.

In addition to the loss of an epitope recognition motif due to global structural rearrangements, locally altered biochemical properties at the direct positions of the respective amino acid residue exchanges must be considered to result in different antibody binding behaviors of SAP-carrying epitope peptides. Of particular interest in our study is the R186Q exchange, through which the hcTn I epitope peptide 2 differs from the wild-type peptide 1. The argininyl residue has been found to be the most mutable residue, and mutations frequently gave rise to non-synonymous SNPs [50]. The Q186 carrying epitope peptide 2 binds stronger to the anti-hcTn I antibody as compared to the affinity of the wild-type epitope peptide 1, despite the fact that this amino acid exchange is located outside the reported epitope region. This increase in binding strength might be explained by reduced steric hindrance and/or a lack of charge repulsion of the Q186 residue with a respective amino acid residue on the antibody´s paratope as compared to the R186 residue of the wild-type epitope peptide. A similar reasoning has been reported for the SARS-CoV-2 omicron spike protein receptor binding domain interaction with human angiotensin-converting enzyme receptor 2, where a Q493R exchange had taken place on the spike protein´s receptor binding domain when compared to the wild-type (Wuhan) strain [51]. Of similar interest are the R192H (peptide 3), R192L (peptide 4), R192C (peptide 5), and R192P (peptide 7) exchanges. The biochemical properties of the argininyl side chain functional guanidinium group cannot be completely mimicked by the side chain atoms of the exchanged amino acid residues, whereas the aliphatic properties of the argininyl side chain stem might be mimicked by leucinyl or iso-leucinyl residues [52]. Hence, non-covalent bonding interactions of R192, such as salt bridges or other polar interactions with an opposing amino acid residue on the antibody´s paratope, which request the presence of the argininyl residue at its defined position on the epitope peptide, are lost in the exchanges that were studied here. Interestingly, while peptides 3, 4, and 5 have been found to bind to the anti-hcTn I antibody still rather strongly in an orthodox fashion, peptide 7 binds only weakly and in an unorthodox fashion to the anti-hcTn I antibody. This comparison strengthens the view that with peptide 7, in addition to the local biochemical changes upon the R192P exchange, greater structural changes have taken place in concert.

SNPs are distributed in the human genome in a non-random fashion [53], and double mutations are considered secondary events in the context of an already existing SAP, as is exemplified with the Andersen-Tawil syndrome, where two missense mutations in the KCNJ2 gene are located on the same allele [54]. While, because of their sparsity, double point mutations seem of lesser importance than SAPs, double point mutations that yield amino acid residue exchanges at neighboring sequence positions have been reported, e.g., as the cause of congenital hypomyelinating neuropathy [55]. Moreover, as was reported for the PCL-γ1 protein, double point mutations at consecutive amino acid residues in the protein alter post-translational modifications as well as associated protein functions [56]. The mass spectrometric ITEM methodology developed by us is capable of investigating altered antibody binding caused by post-translational modification of neighboring amino acid residues within the antibody’s recognition motif [57].

Exchange of single amino acid residues within an epitope [58] or alteration of one epitope´s amino acid side chain, e.g., by post-translational modifications [19], can cause complete loss of molecular recognition or drastically reduce the strong binding of an antibody to its antigen, thereby possibly leading to unorthodox epitope-paratope interactions. On the other hand, an antibody may tolerate exchanges of amino acid residues that, despite falling within an epitope´s core, are not part of the binding motif but are only passively brought into closer contact with the antibody´s paratope surface [17].

By investigating a diagnostically relevant hcTn I epitope and its SNP-caused SAPs, we chose to tackle a medically relevant but hitherto unsolved problem of current antibody-based point-of-care (POC) diagnostics [11]. For “next generation” diagnostic kits, it seems advised to interrogate mutual SAP-related interferences on antibody binding to answer the question whether or not the diagnostic antibody´s capability of abundance difference determination of a selected marker protein is in fact reliable in every real-world scenario. If not, it may be requested to accompany a POC assay’s result with an additional SNP analysis of the individual patient, e.g., through commercial SNP analysis suppliers [59,60], to exclude potential SAP-related diagnostic biases. Alternatively, updated diagnostic kits might have to be equipped with more than one highly specific antibody to cover at least the most abundant SAPs of relevance within a respective antigen´s epitope region of interest to cope with modern precision medicine and individualized/personalized medicine requests [61,62].

## 5. Conclusions

To examine the assumedly serious effects of SAPs on life-threatening conditions that derive from a diagnostic antibody´s recognition disability, a rapid and robust method is desired by which binding strengths between wild-type and mutated epitope peptides and an antibody of concern can be fine-mapped with amino acid residue resolution. Immune complex dissociation in the gas phase, as determined by ITEM-FOUR is a fast and sensitive method by which binding strengths are determined without interference from possible in-solution conformational changes that may occur with the epitope peptide or the respective antigen prior to or during antibody binding. The best correlations between gas-phase binding strengths and in-solution binding affinities occur when epitope peptide conformations remain stable and epitope peptides adopt conformations that are close to the bound conformation. The combination of molecular dynamics simulations with ITEM-FOUR analyses, as shown in this study, allows (i) to detect those epitope peptide ligands for which good correlations between gas phase and in-solution binding strengths are expected, as well as (ii) to predict those ligands for which possible deviations in rankings between gas phase and solution need to be taken into account.

## Figures and Tables

**Figure 1 biomolecules-13-00187-f001:**
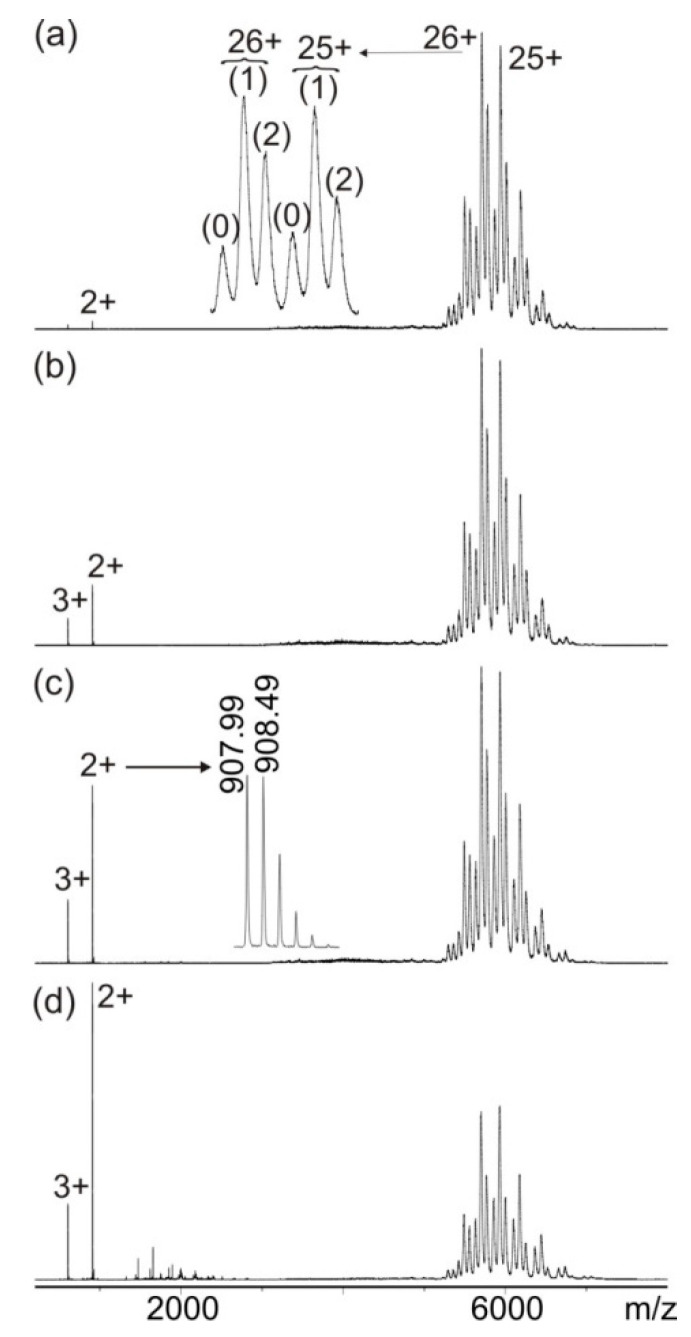
Nano-ESI mass spectra of peptide 1 (ENREVGDWRKNIDAL)—anti-hcTn I antibody immune complex with increasing collision cell voltage differences (∆CV): (**a**) 4 V, (**b**) 16 V, (**c**) 30 V, and (**d**) 80 V. Charge states are given for the immune complexes (right ion series), and the inlet in (**a**) shows a zoom of the 25+ and 26+ ion signals of the antibody (0) and the immune complexes (antibody plus one peptide (1) and antibody plus two peptides (2)). Charge states and m/z values for peptide ion signals are given on the left, and the inlet in (**c**) shows a zoom of the isotopically resolved doubly protonated peptide ion signal. Antibody fragment ion signals are visible between m/z 1200 and m/z 2300 in (**d**). Solvent: 200 mM ammonium acetate, pH 6.7.

**Figure 2 biomolecules-13-00187-f002:**
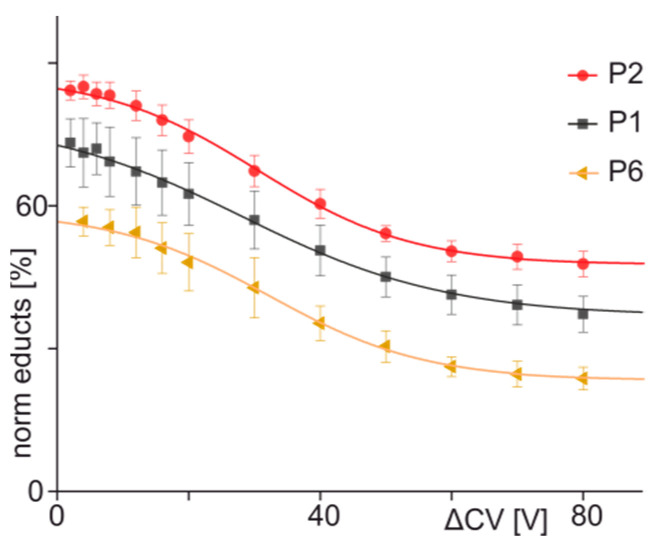
Courses of normalized educt ion intensities of immune complexes of anti-hcTn I antibody and peptides 1 (black squares), 2 (red circles), and 6 (orange triangles) are shown as functions of collision cell voltage differences (∆CV). Each data point is the mean of at least two independent measurements (see Appendix A). Vertical bars indicate standard deviations. The sigmoidal-shaped curves were fitted using a Boltzmann function. The curve parameters are given in Table 2.

**Figure 3 biomolecules-13-00187-f003:**
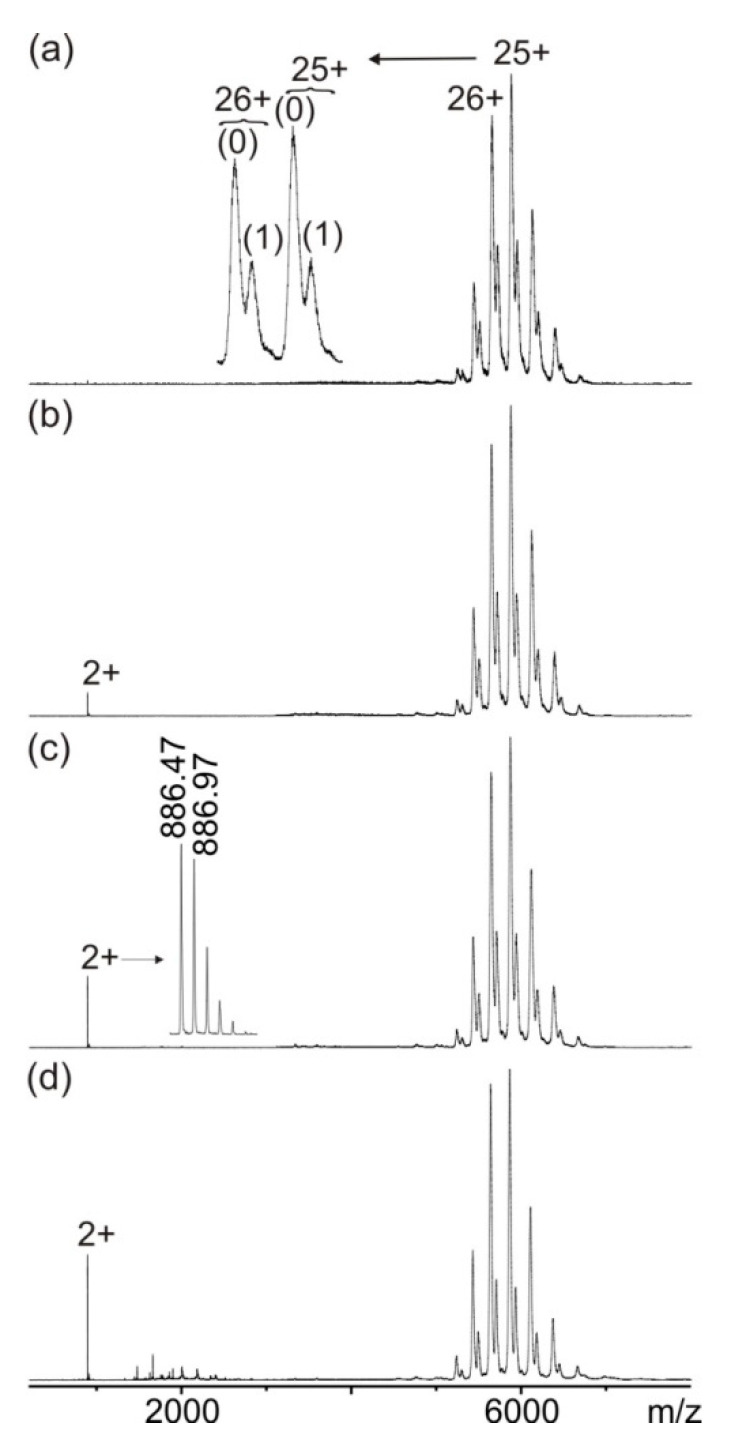
Nano-ESI mass spectra of peptide 4 (ENREVGDWLKNIDAL)—anti-hcTn I antibody immune complex with increasing collision cell voltage differences (∆CV): (**a**) 4 V, (**b**) 16 V, (**c**) 30 V, and (**d**) 80 V. Charge states are given for the immune complexes (right ion series), and the inlet in (**a**) shows a zoom of the 25+ and 26+ ion signals of the antibody (0) and the immune complexes (antibody plus one peptide (1) and antibody plus two peptides (2)). Charge states and m/z values for peptide ion signals are given on the left, and the inlet in (**c**) shows a zoom of the isotopically resolved doubly protonated peptide ion signal. Antibody fragment ion signals are visible between m/z 1200 and m/z 2300 in (**d**). Solvent: 200 mM ammonium acetate, pH 6.7.

**Figure 4 biomolecules-13-00187-f004:**
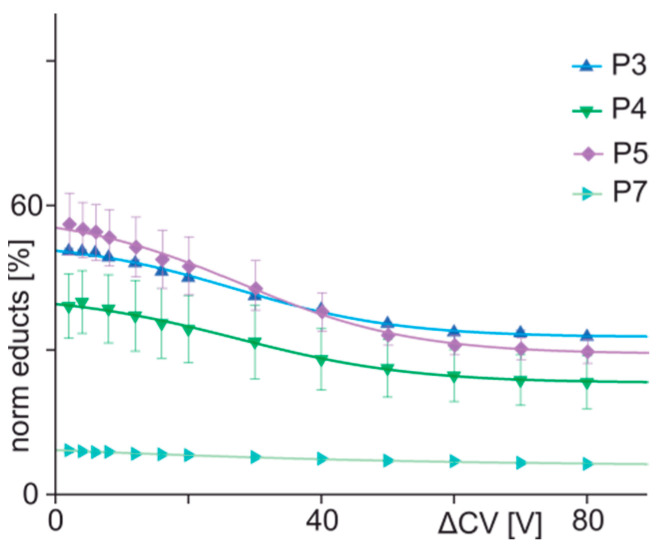
Courses of normalized educt ion intensities of immune complexes of anti-hcTn I antibody and peptides 3 (blue triangles), 4 (green triangles), 5 (purple squares), and 7 (light blue triangles) are shown as functions of collision cell voltage differences (∆CV). Each data point is the mean of at least two independent measurements (see Appendix A). Vertical bars indicate standard deviations. The sigmoidal-shaped curves were fitted using a Boltzmann function. The curve parameters are given in Table 2.

**Figure 5 biomolecules-13-00187-f005:**
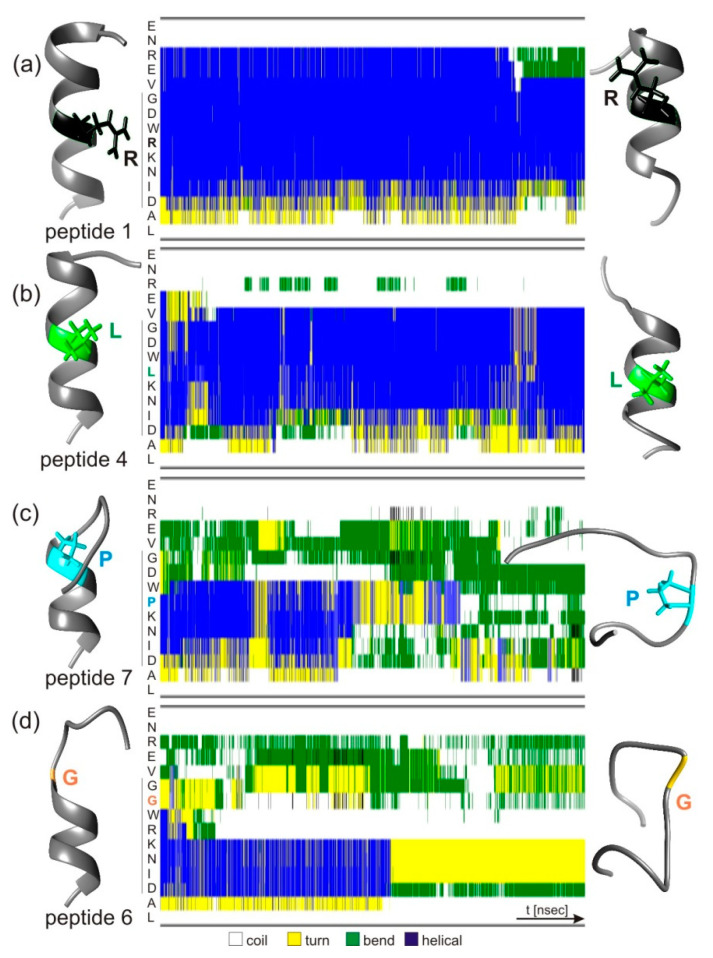
Peptide structure models and secondary structure elements prior to and after molecular dynamics simulations Structure models predicted for epitope peptides (left) were compared to structure models after 50 ns simulations (right). (**a**) peptide 1, (**b**) peptide 4, (**c**) peptide 7, and (**d**) peptide 6. Amino acid residues of peptides are listed from top to bottom (center). The vertical line at the left indicates the epitope region. The secondary structure element in which each residue is involved at a given simulation time point is depicted from left to right as a color-coded bar (10,000 bars per line). Color code: white: coil; yellow: turn; green: bend; blue: helical.

**Table 3 biomolecules-13-00187-t003:** Apparent kinetic and apparent thermodynamic values for gas-phase immune complex dissociations of hcTn I epitope peptides and anti-hcTn I antibodies.

Peptide No.	Amino Acid Sequence ^(a)^	$k_{D\;m0g}^{\#}$ [1/s] ^(b)^	$K_{D\;m0g}^{\#}$ [Ø] ^(b,c)^	$\Delta G_{m0g}^{\#}$ [kJ/mol] ^(b)^	$\Delta H_{m0g}^{\#}$ [kJ/mol] ^(b)^	$T_{amb}\Delta S_{m0g\;}^{\#}$ [kJ/mol] ^(b,d)^
1	ENREVGDWRKNIDAL	1.87 × 10^12^	4.55 × 10^−12^	64.70	−1.91	−66.60
**2**	EN**Q**EVGDWRKNIDAL	7.83 × 10^11^	4.41 × 10^−12^	64.78	−1.57	−66.33
**3**	ENREVGDW**H**KNIDAL	5.09 × 10^12^	4.71 × 10^−12^	64.62	−2.21	−66.82
**4**	ENREVGDW**L**KNIDAL	7.37 × 10^12^	4.77 × 10^−12^	64.59	−2.25	−66.82
**5**	ENREVGDW**C**KNIDAL	3.89 × 10^12^	4.67 × 10^−12^	64.64	−2.04	−66.67
**6**	ENREVG**G**WRKNIDAL	2.90 × 10^12^	4.62 × 10^−12^	64.67	−1.85	−66.50
**7**	ENREVGDW**P**KNIDAL	n.d.	n.d.	n.d.	n.d.	n.d.
**8**	ENREVGDW**P****E**NIDAL	n.d.	n.d.	n.d.	n.d.	n.d.

^(a)^ aa184–aa198 from hcTn I (UniProt: P19429); the epitope region of the monoclonal anti-hcTn I antibody (clone MF4, ab38210 from Abcam) is underlined; amino acid exchanges in peptides 2–8 are printed in bold and are colored. ^(b)^ n.d.: not determined. ^(c)^ unitless number. ^(d)^ T_amb_: 298 K.

**Table 4 biomolecules-13-00187-t004:** Thermodynamic values for immune complex formations of hcTn I epitope peptides and anti-hcTn I antibodies in solution.

Peptide No.	Amino Acid Sequence ^(a)^	$K_{D}^{}$ [Ø] ^(b)^	$K_{A}^{}$ [Ø] ^(b)^	∆G [kJ/mol]	∆H [kJ/mol]	T∆S [kJ/mol] ^(c)^
1	ENREVGDWRKNIDAL	2.19 × 10^−7^	4.58 × 10^6^	−38.05	−60.40	−22.35
**2**	EN**Q**EVGDWRKNIDAL	0.36 × 10^−7^	27.9 × 10^6^	−42.59	−109.10	−66.51
**3**	ENREVGDW**H**KNIDAL	n.d.	n.d.	n.d.	n.d.	n.d.
**4**	ENREVGDW**L**KNIDAL	2.43 × 10^−7^	4.12 × 10^6^	−37.77	−52.10	−14.33
**5**	ENREVGDW**C**KNIDAL	n.d.	n.d.	n.d.	n.d.	n.d.
**6**	ENREVG**G**WRKNIDAL	4.11 × 10^−7^	2.43 × 10^6^	−36.49	−45.40	−8.91
**7**	ENREVGDW**P**KNIDAL	n.b.	n.b.	n.b.	n.b.	n.b.
**8**	ENREVGDW**P****E**NIDAL	n.b.	n.b.	n.b.	n.b.	n.b.

^(a)^ aa184–aa198 from Tn I (UniProt: P19429); the epitope region of the monoclonal anti-hcTroponin I antibody (clone MF4, ab38210 from Abcam) is underlined; amino acid exchanges in peptides 2–8 are printed in bold and are colored. ^(b)^ unitless number; n.d.: not determined; n.b.: not binding. ^(c)^ T = 298 K.

## Data Availability

The mass spectrometry proteomics data have been deposited to the ProteomeXchange Consortium via the PRIDE [33] partner repository with the dataset identifier PXD038365.

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
