# Peer review of "Intact Transition Epitope Mapping—Force Differences between Original and Unusual Residues (ITEM-FOUR)"

_biomolecules, 2023, doi:10.3390/biom13010187_

Round 1

Reviewer 1 Report

It is interesting that the authors discuss the importance of single-point mutation of the peptides and showed the comparison studies with the help of ITEM-FOUR experimental approach. 

On the other hand, the authors should discuss the significance of the double-point mutation of the peptides and compare them with the single-point mutation of the peptides on the basis of the ITEM-FOUR experimental approach. This will highlight the degree of pitfalls approaches for Antibody-based point-of-care diagnostics.

Author Response

Reviewer 1:

It is interesting that the authors discuss the importance of single-point mutation of the peptides and showed the comparison studies with the help of ITEM-FOUR experimental approach.

Comment 1.1:

On the other hand, the authors should discuss the significance of the double-point mutation of the peptides and compare them with the single-point mutation of the peptides on the basis of the ITEM-FOUR experimental approach. This will highlight the degree of pitfalls approaches for Antibody-based point-of-care diagnostics.

Response to comment 1.1:

SNPs are distributed in the human genome in a non-random fashion [Amos, 2010] and double mutations are considered secondary events in the context of an already existing SAP, as is exemplified with the Andersen-Tawil syndrome where two missense mutations in the KCNJ2 gene are located on the same allele [Fukumura et al., 2019]. While because of sparsity double point mutations seem of lesser importance than SAPs, double point mutations which yield in amino acid residue exchanges at neighboring sequence positions have been reported, e.g. as the cause of congenital hypomyelinating neuropathy [Farraj et al., 2006]. Moreover, as was reported for the PCL-γ1 protein, double point mutations at consecutive amino acid residues in the protein alter post-translational modifications as well as associated protein functions [Chung et al., 2010]. The by us developed mass spectrometric ITEM methodology is capable to investigate altered antibody binding upon post-translational modification of neighboring amino acid residues which fall within the antibody´s recognition motif [Scherf et al., 2023]. This information has now been added to the discussion (see p. 15 of the revised manuscript)

Reviewer 2 Report

In this work, Röwer and coworkers describe ITEM-FOUR, an interesting Mass Spectrometry approach for the analysis of SAP effects in epitope-antibody binding. The study is focused on the anti-human cardiac Troponin I in complex with a 15-residues peptide, which is the epitope on the parent protein, plus seven mutated derivatives containing SAPs that have been found to be associated with cardiomyopathies. The work is explicative and detailed, but some comments need to be addressed.

As appears from the results, in the epitope sequence the R in position 9 seems to be crucial for the binding of the wild-type peptide, while the deletion of R in position 3 causes an improvement in the KD, both in MS and in ITC. This suggests that pH might be decisive. In MD simulations it needs to be clarified if the protonation states of the peptides have been built at a particular pH (maybe the one in PBS to mimic the ITC tests in solution?) or at their pI.

Moreover, it would be interesting to see a RMSD plot of the backbone along the MD run to confirm that the system is stabilized since the simulation seems to be a little short for Peptide 1, which toward the end of the run switches to a bend secondary structure in its N-terminus.

Nevertheless, as also reported by the authors in the discussion, the residues 184-198 in Troponin I are part of an intrinsically disordered region in the protein (PDB ID: 1J1E e Alphafold AF-P19429-F1), so the helix structure is not necessary for the binding. Considering also the helix sampling obtained by PepFold3 there seems to be a bias kept for the entire simulation, which is not long enough (and there are no replicas for it) to overcome it. A sampling of extended conformations of the peptides would have been of help. However, I suggest focusing the results and the discussion not on the peptide’s secondary structure but rather on the solvent accessibility of every single residue.

In the discussion, I suggest reporting a short biochemical description of the remarkable results obtained from the bindings of Peptides 2 and 7 with respect to the wt.

The manuscript requests a thorough review of the English form. Finally, as a minor comment, all “ul” and “ml” should be changed in “uL” and “mL” in section 2.1.

Author Response

Reviewer 2:

In this work, Röwer and coworkers describe ITEM-FOUR, an interesting Mass Spectrometry approach for the analysis of SAP effects in epitope-antibody binding. The study is focused on the anti-human cardiac Troponin I in complex with a 15-residues peptide, which is the epitope on the parent protein, plus seven mutated derivatives containing SAPs that have been found to be associated with cardiomyopathies. The work is explicative and detailed, but some comments need to be addressed.

Comment 2.1:

As appears from the results, in the epitope sequence the R in position 9 seems to be crucial for the binding of the wild-type peptide, while the deletion of R in position 3 causes an improvement in the KD, both in MS and in ITC. This suggests that pH might be decisive. In MD simulations it needs to be clarified if the protonation states of the peptides have been built at a particular pH (maybe the one in PBS to mimic the ITC tests in solution?) or at their pI.

Response to comment 2.1:

The protonation states of the simulated peptides were chosen based on their pIs and the pH which had been used in the ITC measurements. The individual protonation states which were used in the simulations are now added to the supplement (see new Supplemental Table 27). The respective information is now mentioned on p. 5 of the revised manuscript.

Comment 2.2:

Moreover, it would be interesting to see a RMSD plot of the backbone along the MD run to confirm that the system is stabilized since the simulation seems to be a little short for Peptide 1, which toward the end of the run switches to a bend secondary structure in its N-terminus.

Response to comment 2.2:

The RMSD plot for peptides 1 and 6 were calculated from the extended 200 ns simulation. Because the RMSD values are influenced by the structure fitting prior to RMSD calculations, we chose to calculate the RMSD of distances between backbone atoms. It can be seen that peptide 1 has a low RMSD (below 0.2 nm) throughout the entire simulation time period whereas peptide 6 displays much larger deviations from the starting structure which is due to unfolding. This information is now shown in the revised supplement (see new Figure S22(b)) and the respective information is now mentioned on pp 5 and 12 of the revised manuscript.

Comment 2.3:

Nevertheless, as also reported by the authors in the discussion, the residues 184-198 in Troponin I are part of an intrinsically disordered region in the protein (PDB ID: 1J1E e Alphafold AF-P19429-F1), so the helix structure is not necessary for the binding. Considering also the helix sampling obtained by PepFold3 there seems to be a bias kept for the entire simulation, which is not long enough (and there are no replicas for it) to overcome it. A sampling of extended conformations of the peptides would have been of help. However, I suggest focusing the results and the discussion not on the peptide’s secondary structure but rather on the solvent accessibility of every single residue.

Response to comment 2.3:

It should be noted that the conformation of the C-terminal partial amino acid sequence 184-198 of hcTn I is not fully established by experiment. In the 1J1E.pdb structure file the partial sequence 192-198 was not resolved whereas the partial sequence 184-191 is partially helical in the intact protein. Alphafold, being a state-of-the-art protein structure prediction system, assumes an unstructured partial sequence for the respective C-terminal partial sequence. However, one ought to keep in mind the limitations of Alphafold which cannot yet deal successfully with multimeric proteins and/or protein complexes. Troponin is a protein complex which is composed of three subunits (Tn I, Tn T, and Tn C). Also, Alphafold does not predict positions of metal ions, like Ca2+ in troponin, cofactors, and other ligands in protein structures [Perrakis et al., 2021]. Moreover, it is not known how the Alphafold artificial intelligence prediction engine handles short stretches of amino acid residues when extracted from their native/intact protein environment. Therefore, we regard Alphafold structure predictions for C-terminal troponin I epitope peptides with caution. On the other hand Pepfold3 is a system and molecular dynamics simulations software package which has been successfully used in the past for structure predictions of short peptides [Lamiable et al., 2016; Geng et al., 2019]. The philosophy behind the computational study which is presented in our work was to first use the Pepfold3 system for an initial prediction of the in-solution conformations of all studied epitope peptides and next to use molecular dynamics to comparatively evaluate the stabilities of the predicted conformations using the same time intervals (50 ns) for all peptides.

The reviewer's comment on extending the simulation time period is indeed valuable and thus we repeated and extended the simulation time to 200 ns for epitope peptides 1 and 6. The secondary structure calculations for these extended simulations clearly showed that peptide 1 remained helical while peptide 6 lost its partial helical conformation. The extended simulations confirm the structure analyses which were already obtained with the initial shorter simulation time periods of 50 nsec, each.

This information is now shown in the revised supplement (see new Figure S22(a)) and the respective information is now mentioned on pp 12/13 of the revised manuscript.

Concerning the solvent accessible surface areas (SASAs) of each residue of each of the epitope peptides we find the suggestion made by the reviewer very helpful and we calculated the SASAs for each residue of all the epitope peptides for both, the epitope peptide structures prior to and after simulation (50 nsec). While one can see changes of SASAs in these comparisons for all peptides, one can qualitatively distinguish epitope peptides 1, 2, and 4 with rather little changes from epitope peptides 3, 5, 6, 7, and 8 with much larger changes of SASAs. These results are in line with the information which is e.g. obtained by the RMSD calculations and agree with keeping the alpha helical structure by peptides 1, 2, and 4 and rather rapid loss of alpha helical structures by peptides 3, 5, 6, 7, and 8. The individual SASA pairs (before and after simulation) and the respective SASA differences are now added to the supplement (see new Supplemental Tables 28 to 35). The respective information is now mentioned on pp 5 and 12/13 and of the revised manuscript.

Comment 2.4:

In the discussion, I suggest reporting a short biochemical description of the remarkable results obtained from the bindings of Peptides 2 and 7 with respect to the wt.

Response to comment 2.4:

In addition to induced more global structural rearrangements locally altered biochemical properties at the direct positions of the respective amino acid residue exchanges must be considered to also play decisive roles for differences in binding behaviors of SAP-carrying epitope peptides. Of particular interest in our examples is the R186Q exchange through which the hcTn I epitope peptide 2 differs from wild type peptide 1. The argininyl residue has been found to be the most mutable residue; and mutations frequently gave rise to non-synonymous SNPs [de Beer et al., 2013]. The Q186 carrying epitope peptide 2 binds stronger to the anti-hcTn I antibody as compared to the affinity of the wild type epitope peptide 1, despite the fact that this amino acid exchange is located outside the reported epitope region. This increase in binding strength might be explained with reduced steric hindrance and/or lack of charge repulsion of the Q186 residue with a respective amino acid residue on the antibody´s paratope as compared to the R186 residue of the wild type epitope peptide. A similar reasoning has been reported for the SARS-CoV-2 omicron spike protein receptor binding domain interaction with human angiotensin converting enzyme receptor 2 where a Q493R exchange had taken place on the spike protein´s receptor binding domain when compared to the wild-type (Wuhan) strain [Glocker et al., 2022]. Also of interest are the R192H (peptide 3), R192L (peptide 4), R192C (peptide 5), and R192P (peptide 7) exchanges. The biochemical properties of the argininyl side chain functional guanidinium group cannot completely be mimicked by the side chain atoms of the exchanged amino acid residues whereas the aliphatic properties of the argininyl side chain stem might be mimicked by leucinyl or iso-leucinyl residues [Betts and Russel, 2003]. Hence, non-covalent bonding interactions of R192, such as salt bridges or other polar interactions with an opposing amino acid residue on the antibody´s paratope which request the presence of the argininyl residue at its defined position on the epitope peptide are lost by the exchanges which were studied here. Interestingly, while peptides 3, 4, and 5 have been found to still rather strongly bind to the anti-hcTn I antibody in an orthodox fashion, peptide 7 binds only weakly and in an unorthodox fashion to the anti-hcTn I antibody. This comparison strengthens the view that with peptide 7 in addition to the local biochemical changes upon the R192P exchange greater structural changes had taken place in concert. This information has now been added to the discussion (see p. 14/15 of the revised manuscript)

Comment 2.5:

The manuscript requests a thorough review of the English form. Finally, as a minor comment, all “ul” and “ml” should be changed in “uL” and “mL” in section 2.1.

Response to comment 2.5:

We thank the reviewer for this comment. The entire text had been edited to fulfill the request. To follow the Journal´s recommendations exchanges of “l” to “L” have been undertaken throughout the manuscript. We apologize for the oversight.

Cited references:

Amos, 2010. Proc. Biol. Sci. Proc Biol Sci., 277, 1443–1449.

Fukumura et al., 2019. J. Neurol. Sci., 407, 116521.

Farraj et al., 2006. Neurobiol. Dis., 24,159-169.

Chung et al., 2010. Exp. Mol. Med., 42, 216-222.

Scherf et al., 2023. J. Am. Soc. Mass Spectrom., in press.

Perrakis et al., 2021. EMBO reports, 22, e54046.

Lamiable et al., 2016. Nucl. Acids Res., 44, W449–W454.

Geng et al., 2019. Comput. Struct. Biotechnol. J., 17, 1162–1170.

de Beer et al., 2013. PLOS Comp. Biol., 91, e1003382.

Glocker et al., 2022. Medicina, 58, 226.

Betts and Russel, 2003. in: Bioinformatics for Geneticists, Eds. M.R. Barnes and I.C. Gray, chapter 14.

Reviewer 3 Report

This is a very well-conducted study by Claudia Röwer and collaborators in which they compared the binding energies in solution and gas phase using a technique developed in their own lab, which they call ITEM-FOUR. ITEM -FOUR uses mass spectrometry and provides a good correlation between the binding energies obtained in both states. In addition, it can differentiate single amino polymorphisms in the association of antibody-peptide complexes. In summary, a nice and relevant work that opens new perspectives in immunology using mass spectrometry.

Author Response

Reviewer 3:

This is a very well-conducted study by Claudia Röwer and collaborators in which they compared the binding energies in solution and gas phase using a technique developed in their own lab, which they call ITEM-FOUR. ITEM -FOUR uses mass spectrometry and provides a good correlation between the binding energies obtained in both states. In addition, it can differentiate single amino polymorphisms in the association of antibody-peptide complexes.

Comment 3.1:

In summary, a nice and relevant work that opens new perspectives in immunology using mass spectrometry.

Response to comment 3.1:

We thank the reviewer for his encouraging comment and are grateful for his review.

Round 2

Reviewer 2 Report

The questions have been addressed and therefore I recommend the paper for its publication.